# Induced Tissue-Specific Stem Cells (iTSCs): Their Generation and Possible Use in Regenerative Medicine

**DOI:** 10.3390/pharmaceutics13060780

**Published:** 2021-05-23

**Authors:** Issei Saitoh, Masahiro Sato, Yuki Kiyokawa, Emi Inada, Yoko Iwase, Natsumi Ibano, Hirofumi Noguchi

**Affiliations:** 1Department of Pediatric Dentistry, Asahi University School of Dentistry, Hozumi 501-0296, Japan; bano@dent.asahi-u.ac.jp; 2National Center for Child Health and Development, Department of Genome Medicine, Tokyo 157-8535, Japan; sato-masa@ncchd.go.jp; 3Division of Pediatric Dentistry, Faculty of Dentistry & Graduate School of Medical and Dental Sciences, Niigata University, Niigata 951-8514, Japan; ykiyokawa@dent.niigata-u.ac.jp; 4Department of Pediatric Dentistry, Kagoshima University Graduate School of Medical and Dental Sciences, Kagoshima 890-8544, Japan; inada@dent.kagoshima-u.ac.jp; 5Department of Dentistry for the Disabled, Asahi University School of Dentistry, Hozumi 501-0296, Japan; iwase@dent.asahi-u.ac.jp; 6Department of Regenerative Medicine, Graduate School of Medicine, University of the Ryukyus, Okinawa 903-0215, Japan; noguchih@med.u-ryukyu.ac.jp

**Keywords:** induced tissue-specific stem cells (iTSCs), induced pluripotent stem cells (iPSCs), partial reprogramming, naïve stem cells, epiblast stem cells, insulin-producing cells

## Abstract

Induced tissue-specific stem cells (iTSCs) are partially reprogrammed cells which have an intermediate state, such as progenitors or stem cells. They originate from the de-differentiation of differentiated somatic cells into pluripotent stem cells, such as induced pluripotent stem cells (iPSCs) and embryonic stem cells (ESCs), or from the differentiation of undifferentiated cells. They show a limited capacity to differentiate and a morphology similar to that of somatic cell stem cells present in tissues, but distinct from that of iPSCs and ESCs. iTSCs can be generally obtained 7 to 10 days after reprogramming of somatic cells with Yamanaka’s factors, and their fibroblast-like morphology remains unaltered. iTSCs can also be obtained directly from iPSCs cultured under conditions allowing cellular differentiation. In this case, to effectively induce iTSCs, additional treatment is required, as exemplified by the conversion of iPSCs into naïve iPSCs. iTSCs can proliferate continuously in vitro, but when transplanted into immunocompromised mice, they fail to generate solid tumors (teratomas), implying loss of tumorigenic potential. The low tendency of iTSCs to elicit tumors is beneficial, especially considering applications for regenerative medicine in humans. Several iTSC types have been identified, including iTS-L, iTS-P, and iTS-D, obtained by reprogramming hepatocytes, pancreatic cells, and deciduous tooth-derived dental pulp cells, respectively. This review provides a brief overview of iPSCs and discusses recent advances in the establishment of iTSCs and their possible applications in regenerative medicine.

## 1. Introduction

Induced pluripotent stem cells (iPSCs) have been successfully generated from a variety of mammalian cells, including patient-derived cells, through the enforced expression of stem cell-specific transcription factors such as POUF5F1 (OCT3/4 or OCT4; hereafter called OCT3/4), sex-determining region Y-box 2 (SOX2), Krüppel-like factor 4 (KLF4), and c-MYC (OSKM) in differentiated somatic cells (e.g., fibroblasts) [1,2] (Figure 1). iPSCs exhibit unlimited self-renewal potential in vitro and the ability to differentiate into various cell types derived from the three embryonic germ layers: mesoderm, ectoderm, and endoderm [3,4,5,6,7]. They are also recognized as promising resources in regenerative medicine because they can be created from somatic cells of the patients themselves, thereby allowing autologous (self)-transplantation [1]. However, the manipulation of the host genome via the integration of viral vector-derived components, needed for the generation of iPSCs, often raises critical safety concerns; for instance, host-genome manipulations may result in insertional mutations that interfere with the normal function of iPSC derivatives [2,3] or even induce tumorigenesis [6,8]. Furthermore, iPSCs often retain a transcriptional memory of the original somatic cell [9], which may occasionally constrain the ability of iPSCs to differentiate into the desired cell type [3]. Therefore, the potential use of iPSCs in regenerative medicine is still limited.

### 1.1. Reprogramming of Somatic Cells into iPSCs

In 2008, the mechanism underlying the reprogramming process of somatic cells into iPSCs was first investigated by Jaenisch et al. [10,11] Later, it was shown that this reprogramming process can be roughly classified into three phases, namely “initiation”, “maturation”, and “stabilization” (as shown in Figure 1) [12,13]. Alkaline phosphatase (ALP) and F-box only protein 15 (FBXO15) are the molecular markers defining the initial steps of the de-differentiation of somatic cells (initiation phase; shown in Figure 1), whereas endogenous transcripts of OCT3/4 and SOX2 are detectable only late during iPSC generation [13]. Notably, single-cell analysis of the expression of 26 genes demonstrated that expression changes occur homogeneously at early (days 0–3; corresponding to the initiation phase) and late time points (day 9 onward; corresponding to the stabilization phase), whereas they are heterogeneous at the intermediate stages (days 6–9; corresponding to the maturation phase) [12].

To date, several reprogramming strategies have been reported. For example, lentivirus-based vectors had a higher reprogramming efficiency [3,14], but they tend to exhibit chromosomal integration into the host genome, like a retrovirus vector. Adenovirus-based vectors [15] and Sendai virus [16,17] are also useful to establish iPSCs as non-integrating viral vectors. Furthermore, the use of *piggy*Bac transposon cassettes carrying reprogramming factors [18,19], the direct delivery of RNA [20], and a chemical approach for reprogramming [21] are reported to effectively establish iPSCs.

Concerning the molecular mechanism of reprogramming, suppression of somatic cell genes occurs, leading to mesenchymal-to-epithelial transition [12,22]. Metabolic changes from oxidative phosphorylation to glycolysis also occur [23,24]. Dr. Abeliovich’s group [25] found that poly (ADP-ribose) polymerase-1 (Parp1) and ten-eleven translocation-2 (Tet2) are both involved in the earliest stages of reprogramming. These two proteins result in modifications to both DNA and histones that allow the reprogramming factors, typified by Oct-3/4, to access to the promoters of target genes. Furthermore, recent work has suggested that a subset of transcription factors (TFs), so-called Pioneer TFs, play an important role during the stochastic phase of iPSC reprogramming [26]. Pioneer TFs bind directly to condensed chromatin, leading to its opening. This chromatin decondensation exposes specific gene promoters in the DNA, to which TFs can directly bind.

### 1.2. Epigenetic Memory

It is well known that DNA methylation, the addition of methyl (CH_3_) groups to DNA molecules, is an epigenetic mechanism that affects gene expression, and therefore, is behind the modification of gene function. In 2010, Hochedlinger’s group first assessed the epigenetic landscape in iPSCs. Briefly, they produced iPSCs derived from mouse fibroblasts, hematopoietic cells, and myogenic cells and explored the potential differences in the transcriptional and epigenetic patterns among the three iPSC populations using a genome-wide, restriction enzyme-based methylation analysis [27]. Interestingly, Polo et al. [27] found a distinct pattern of gene expression among these three types of iPSC. Furthermore, it was shown that the cellular origin influences the in vitro differentiation potential of iPSCs. These data clearly suggest that each iPSC line has a specific memory of the cell of origin. In other words, somatic cell-derived iPSCs cannot completely erase DNA methylation, which may lead to their poor differentiation capacity [27,28]. Of note, the above phenomenon, called “epigenetic memory”, is an essential process of life that governs the inheritance of the predestined functional characteristics of normal cells in the context of all differentiated lineages, as reviewed by Thiagalingam [29].

### 1.3. Potential of Human Embryonic Stem Cells (hESCs)/Human iPSCs (hiPSCs) for Cell-Based Therapies

The discovery of hESCs and of their potential to differentiate into many types of cells [30] led to the hypothesis that hESC-derived cells can potentially be used to replace or restore tissues that have been damaged by disease or injury, in the context of pathologies such as type 1 diabetes (T1D), heart attack, Parkinson’s disease, age-related macular degeneration, or spinal cord injury [31].

hESCs are created from blastocysts that have been derived from in vitro fertilized oocytes (provided by volunteers) [30]; obviously, they are associated with ethical concerns, because the creation of hESCs involves the destruction of human embryos. To bypass this problem, some groups employed a somatic cell nuclear transfer (SCNT), by which the nucleus of a patient’s own somatic cell is placed into the cytoplasm of enucleated oocytes under observation using a micromanipulator system, for the generation of SCNT embryo-derived hPSCs [32,33,34]. Of note, the nuclei of somatic cells is reprogrammed to acquire a totipotent phenotype when they are transferred into the egg cytoplasm via SCNT [35]; this epoch-making process is called “nuclear reprogramming of differentiated cells” or “cloning”. Importantly, because the SCNT-treated offspring are composed of cells with the donor genetic information, cells from these SCNT embryos or fetuses can be used in the context of cell-based self-transplantation. However, this SCNT-based approach has some drawbacks. It still requires human oocytes from volunteers or frozen oocytes that are initially stored for assisted reproductive techniques but are later discarded. Furthermore, in almost all cases, only a limited number of SCNT-derived embryos are available. This may hamper the generation of a high number of hESCs at once.

Because of the discovery of murine iPSCs by Yamanaka’s group in 2006 [1], hiPSCs have been considered a promising alternative to hESCs. As previously mentioned, iPSCs are generated from differentiated somatic cells through reprogramming events. In other words, iPSC generation does not require fertilized oocytes. Therefore, the use of hiPSCs is not associated with major ethical issues [36,37]. Because hiPSCs have totipotent capabilities like hESCs and can circumvent the need for SCNT-treated embryos, they are now recognized as a promising resource suitable for clinical use in regenerative medicine, as reviewed elsewhere [3].

From a clinical point of view, two limitations are always associated with cell-based therapies using hESCs/hiPSCs. The first one is that these cells have a tumorigenic potential; in fact, teratoma formation always occurs after cell transplantation. Of note, teratoma formation can even occur after the transplantation of in vitro pre-differentiated cells, due to the potential residual undifferentiated cells. Additionally, the second limitation is that the available methods for the differentiation of hESC/hiPSCs into different cell types are still limited and have a less-than-optimal efficacy.

## 2. The Discovery of Induced Tissue-Specific Stem Cells (iTSCs), as an Alternative to ESCs/iPCs

Induced tissue-specific stem cells (iTSCs) are defined as intermediate cells (also called progenitors or stem cells) originating from the de-differentiation process of differentiated somatic cells (e.g., fibroblasts) into pluripotent stem cells (e.g., iPSCs and ESCs); they are partially reprogrammed cells, in an intermediate state, just between somatic cells and iPSCs (Figure 1). In 2015, Noguchi et al. [38] first developed a method for the generation of iTSCs in mice. Briefly, primary cultured pancreatic cells from 24-week-old C57BL/6 mice were transfected with a single plasmid encoding OSKM, several times (on days 1, 3, 5, and 7 after seeding). Cobblestone-like colonies (similar to those of mouse pancreatic stem cells, but distinct from iPSC-derived colonies) appeared 30–45 days after the first transfection. Importantly, these colonies could be manually picked and propagated in vitro. The resulting iTSCs (iTSC-P cells) generated from primary pancreatic cells expressed several genetic markers typical of endoderm and pancreatic progenitors. Of note, the subcutaneous transplantation of iTSC-P cells into immunodeficient mice failed to generate teratomas. Remarkably, iTSC-P cells differentiated into insulin-producing cells (IPCs) more efficiently than ESCs [38]. Additionally, Bar-Nur et al. [39] reported that iPSCs derived from pancreatic β-cells tend to differentiate more readily than iPSCs derived from other tissues. According to Bar-Nur et al. [39], this is due to the presence of epigenetic memory as mentioned above. Moreover, Saitoh et al. [40] demonstrated that primary pancreatic cells isolated from non-obese diabetic (NOD) mice (aged 6 months old), an animal model of T1D, can equally generate iTSC-P cells after transfection with an OSKM-expression vector (Figure 2). Of note, the resulting iTSC-P cells proliferated in vitro and expressed pluripotency-related genes (e.g., OCT3/4 and SOX2) as well as the gene coding for pancreatic-duodenal homeobox factor-1 (Pdx1), a transcription factor specific of stem cells for the pancreatic β-cell lineage. Once again, the transplantation of iTSC-P cells into nude mice failed to produce teratomas. Furthermore, iTSC-P cells had the ability to produce insulin in response to glucose when they were induced to differentiate into pancreatic β-cells. Importantly, altogether, these findings suggest the possibility of the production of functional pancreatic β-cells, directly from patients with T1D; therefore, iTSC-P cells may be a safe and effective resource for cell-based therapies in the context of T1D.

### 2.1. Induction of Human iTSCs (hiTSCs)

Bone marrow-derived mesenchymal stem cells and adipose-derived mesenchymal stem cells (ADSCs) are now recognized as useful materials for the treatment of various diseases [41,42,43,44]. However, these cells are sometimes difficult to obtain and maintain in vitro. In this context, iTSCs appear to be convenient, because they proliferate continuously in vitro; therefore, it is easy to acquire large amounts of these cells at once for clinical use.

In general, the generation of iPSCs appears to depend on many factors including the age, type and origin of the primary cells used, and passage number [45]. For example, the difficulty to establish iPSCs from cells isolated from aged donors was previously reported [46]. However, contradictory reports have been provided by Yagi et al. [47]; they successfully generated iPSCs from an older patient at 105 years of age. Conversely, Inada et al. [48] reported similar problems, but in the context of cells from young individuals; when human deciduous tooth-derived dental pulp cells (HDDPCs) isolated from individuals with 6–11 years old were transfected with vectors carrying OSKM, of the five primarily isolated HDDPCs examined, two (called HDDPC-1 and -5) were successfully reprogrammed into iPSCs, whereas the other three cells were not. Notably, HDDPC-1 and -5 exhibited higher levels of expression of ALP, a useful stemness marker (Figure 1); the expression of OCT3/4 was also prominent in these two lines.

In fact, Soda et al. [49] focused on the expression of ALP, observable in the early steps of reprogramming (as shown in Figure 1) to confirm that ALP can be a useful marker for the definition of intermediate cells. Briefly, they examined the expression of ALP in HDDPCs at different timings after transfection with OSKM-expressing vectors. Interestingly, they found increased ALP levels in cells 7 to 10 days after transfection; of note, at these time points, fibroblastic cell morphology was unaltered. Conversely, OSKM-treated HDDPCs at day 10 exhibited molecular stemness properties and multipotency. Furthermore, transplantation of these cells into the pancreatic parenchyma of nude mice resulted in no solid tumor formation, whereas the transplantation of HDDPC-derived iPSCs caused solid tumors (teratoma). These data show that ALP-enriched HDDPCs (obtained 10 days after reprogramming) are intermediate cells. Based on these data, Soda et al. [49] called these cells “hiTSC-D”. As mentioned previously, every differentiated cell has an epigenetic memory, which is specific to each cell type. If this hypothesis is true, it may be possible to isolate intermediate cells from various differentiated cells through transfection using OSKM-expressing vectors. Importantly, the monitoring of ALP expression may be essential to identify the intermediate cells.

Notably, Miyagi-Shiohira et al. [50] succeeded in the generation of human iTSCs from aged ADSCs (called “hiTS-M cells”, with a restricted self-renewal capacity), using a single, synthetic, self-replicating Venezuelan Equine Encephalitis reprogramming factor RNA replicon (termed “SR-RNA vector”) expressing OSK and GLIS family zinc finger 1 (GLIS1). Of note, this SR-RNA vector was previously used to generate iPSCs. Importantly, hiTS-M cells survived for 15 passages and expressed cell surface markers similar to those of human ADSCs (hADSCs). Furthermore, they were able to differentiate into fat cells and osteoblasts after treatment with differentiation-inducing drugs.

### 2.2. Induction of hiTSCs from hiPSCs

It was reported that there are at least two types among mouse ESCs in 2007, namely the naïve and primed cells, called naïve stem cells (NSCs) and epiblast stem cells (EpiSCs), respectively [51,52]. The colony morphology, X inactivation status in female cells, and growth factor requirements for the maintenance of the pluripotent state, are remarkable differences between the two cell types [53,54]. Of note, it is thought that hESCs are more closely related to mouse EpiSCs [55]. However, hiPSCs, which are generated via factor-based reprogramming of adult somatic cells, are thought to be classified as EpiSCs; they have unlimited self-renewal potential and differentiate into three germ layers in vitro but have limited in vivo pluripotency [10], suggesting that they in fact are not pluripotent. Of note, Choi et al. [56] demonstrated that the established EpiSCs retain an epigenetic memory of the tissue of origin, which may influence efforts at directed differentiation. Interestingly, Kim et al. [28] suggested that SCNT is more effective at establishing the ground state of pluripotency than factor-based reprogramming for applications in disease modeling or treatment. However, NSCs may be convenient to obtain by chemical induction, due to their simplicity and the use of pre-existing EpiSCs [57,58,59,60,61]. For example, it was reported that NSCs could be successfully induced by reprogramming-related drugs via the treatment of human iPSCs with 2i (PD0325901; an inhibitor of the MEK/ERK pathway + CHIR99021; an activator of the Wnt/β-catenin pathway), kenpaullone (an inhibitor of glycogen synthase kinase 3), or forskolin (known to increase the intracellular levels of cyclic AMP) [62]. Importantly, the resulting NSC-like cells had the same features of murine ESCs in their dome-like colony morphology, a higher survival rate after trypsinization (indicating a lower dependency on the Rho-associated coiled-coil forming kinase inhibitor), rapid proliferation, and activation of both X-chromosomes in female NSCs. Of note, these findings suggest that NSCs might have lost their specific epigenetic memory and are in a state of pluripotency. Additionally, using the method reported by Hanna et al. [62], Inada et al. [63] recently demonstrated that HDDPC-derived iPSCs (hereafter referred to as HDDPC-EpiSCs) could be successfully converted into NSC-like cells (hereafter referred to as HDDPC-NSCs) with specific properties such as accelerated growth rate, dome-like colony morphology, expression of ESC-associated stem cell-specific markers (e.g., reduced expression protein-1 (REX-1), also called zinc finger protein 42 (ZFP-42)), expression of stage-specific embryonic antigen 1 (SSEA-1), and the ability to differentiate into mouse vasa homolog-positive cells after teratoma formation in vivo. In contrast, HDDPC-EpiSCs exhibited poorer differentiation potency than NSCs in the context of teratoma formation in vivo.

### 2.3. Induction of IPCs from HDDPC-NSCs

In the results of Inada et al. [63], HDDPC-NSCs lost their epigenetic memory. Then, Kiyokawa et al. [64] reported that HDDPC-NSCs were more readily induced to differentiate into pancreatic β-cells than HDDPC-EpiSCs, as depicted in Figure 3.

They produced embryoid bodies (EBs) from HDDPC-NSCs via incubation in non-coated plates to initiate early differentiation and then transferred them into coated plates to accelerate cell spreading. After cell attachment, these cells were treated with factors to promote their differentiation into the pancreatic β-cell lineage. As a control, HDDPC-EpiSCs were subjected to the same interventions. Importantly, HDDPC-NSCs efficiently generated intermediate cells that expressed PDX1 and were capable of producing insulin, whereas HDDPC-EpiSCs did not (Figure 4). Additionally, the intrapancreatic grafting of in vitro-formed β-cells into nude mice generated a cell mass containing IPCs, without noticeable tumorigenesis (Figure 5). Altogether, these data suggest that HDDPC-NSCs can be used as a promising source to cure T1D.

## 3. Conclusions

The transient OSKM overexpression is able to reprogram all types of somatic cells. In the early phase of reprogramming, suppression of somatic cell genes occurs, leading to mesenchymal-to-epithelial transition [12,22]. Metabolic changes from oxidative phosphorylation to glycolysis also occur [23,24]. Expression of Parp1 and Tet2 causes modifications of DNA and histones, allowing the reprogramming factors to access to the promoters of target genes [25]. Chromatin decondensation caused by expression of Pioneer TFs may also allow exposure of specific gene promoters in the DNA, to which TFs can directly bind [26]. Conversely, the late phase of reprogramming includes the activation of pluripotency-associated genes, repression of tissue-specific TFs and developmental genes, and bivalent methylation of H3K4me3 and H3K27me3 in high CpG promoter regions. In other words, for successful reprogramming leading to the generation of iPSCs, many events may occur sequentially or in parallel.

It is conceivable that molecular events related to the early phase of reprogramming first occur after transfection with reprogramming inducing factors, and then iTSCs are formed during the conversion of somatic cells into iPSCs. iTSCs are tissue-specific stem cells, as exemplified by their multipotentiality, expression of stemness-specific molecules, and non-tumorigenic potential. Importantly, they can be produced from iPSC-derived NGSs. Of note, iTSCs can be further treated to achieve terminal differentiation and produce pancreatic β-cells with the potential to produce insulin. Because iTSCs can be induced from other histiocytes, it may be possible to produce and store many types of iTSCs. These results suggest that iTSCs can serve as valuable resources for cell-based therapies.

## Figures and Tables

**Figure 1 pharmaceutics-13-00780-f001:**
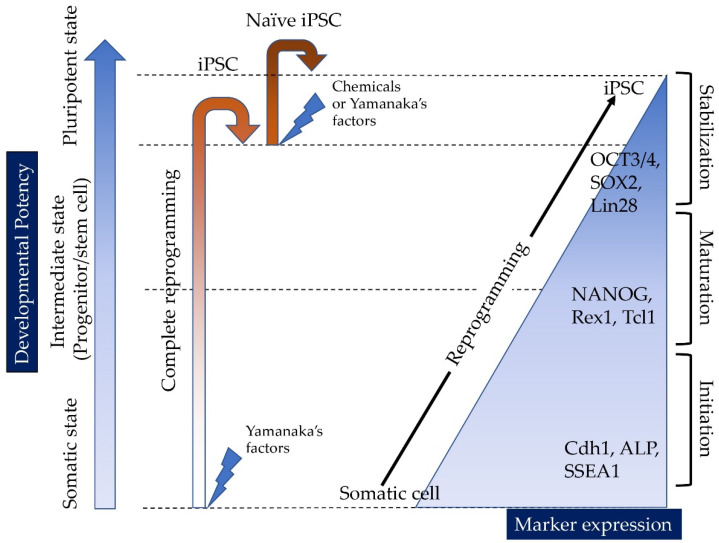
Cell state and molecular events during the reprogramming of somatic cells into iPSCs. When complete reprogramming occurs, somatic cells are successfully converted into iPSCs. The resulting iPSCs can be further reprogrammed into naïve iPSCs through transfection with vectors carrying Yamanaka’s factors or via treatment with chemicals. There are at least three phases with respect to (de-)differentiation (“somatic state”, which may correspond to the initiation stage, “intermediate state”, which may correspond to the maturation stage, and “pluripotent state”, which may correspond to the stabilization stage), according to Samavarchi-Tehrani et al. [12]. Importantly, several molecular markers define each of the above phases, according to Samavarchi-Tehrani et al. [12] and Polo et al. [13]. CDH1, E-cadherin; ALP, alkaline phosphatase; SSEA-1, stage-specific embryonic antigen 1; NANOG, nanog homeobox; REX1, reduced expression protein-1; TCL1, T cell lymphoma breakpoint 1; OCT3/4, octamer-binding transcription factor-3/4; SOX2, sex-determining region Y-box 2; LIN28, lin-28.

**Figure 2 pharmaceutics-13-00780-f002:**
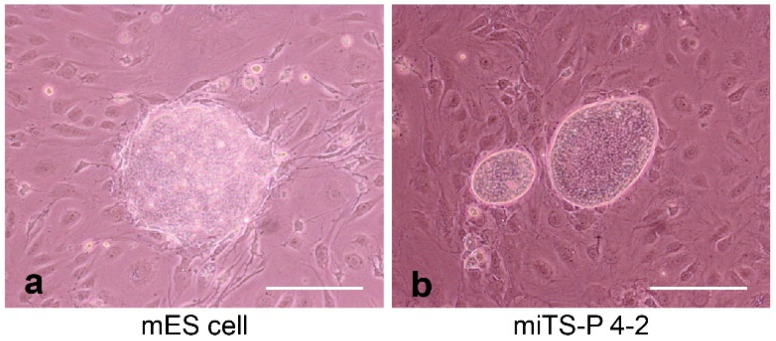
iTSCs (iTSC-P cells) generated from primary pancreatic cells isolated from non-obese diabetic (NOD) mice after transfection with an OSKM-expression vector. The resulting iTSC-P cells (miTS-P 4-2 in b) exhibit a cobblestone-like morphology, different from that of the mouse ESCs’ mole-like morphology (mES cells in a). We obtained these images from experiments performed exclusively for this review. Scale = 200 μm.

**Figure 3 pharmaceutics-13-00780-f003:**
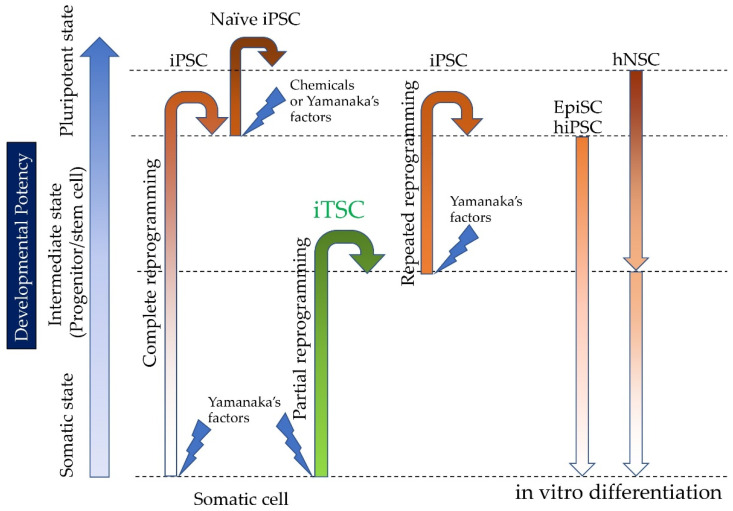
Somatic cells are converted into intermediate cells called iTSCs, when partial reprogramming occurs. These iTSCs can be further reprogrammed to form iPSCs via repeated transfection with vectors carrying Yamanaka’s factors. Epi-stem cells (EpiSCs; corresponding to human iPSCs) differentiate poorly upon induction, whereas naïve stem cells (NSCs; corresponding to mouse iPSCs and ESCs) do not. Because NSCs are thought to lose their epigenetic memory, they may be more efficiently induced to differentiate into various cell types (versus EpiSCs).

**Figure 4 pharmaceutics-13-00780-f004:**
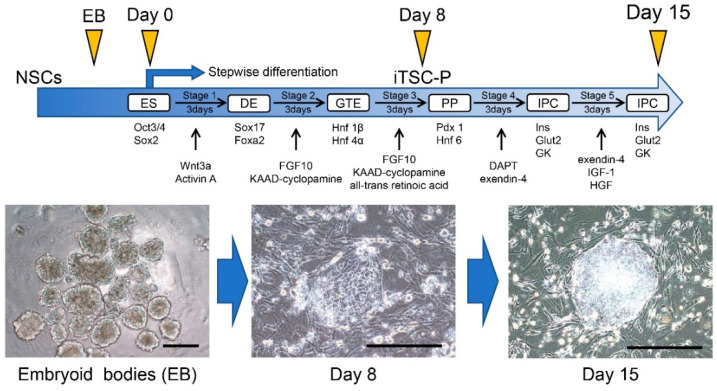
The new protocol enabling the differentiation of HDDPC-NSCs into pancreatic β-cells. The schedule for the in vitro differentiation of HDDPC-NSCs, shown in the upper panel, is adapted from Kiyokawa et al. [64], MDPI, 2020. After formation of embryoid bodies (EB) from HDDPC-NSCs, the resultant EBs were seeded onto a tissue-culture dish to them to promote outgrowth for 2 days. Next, these cells were subjected to a stepwise protocol to drive differentiation toward IPCs (upper panel). In Stage 1, the cells were treated with Wnt3a and activin A for 1 day, followed by treatment with activin A + 0.2% FBS for 2 days. In Stage 2, the cells were treated with fibroblast growth factor 10 (FGF10) and 3-keto-*N*-(aminoethyl-aminocaproyl-dihydrocinnamoyl) cyclopamine (KAAD-cyclopamine) + 2% FBS for 3 days. In Stage 3, the cells were treated with FGF10, KAAD-cyclopamine, and all-trans retinoic acid + 1% (vol/vol) B27 supplement for 3 days. In Stage 4, the cells were treated with *N*-[*N*-(3,5-difluorophenacetyl)-l-alanyl]-*S*-phenylglycine t-butyl ester (DAPT) and exendin-4 + 1% (vol/vol) B27 supplement for 3 days. In Stage 5, the cells were treated with exendin-4, insulin-like growth factor 1 (IGF-1), and hepatocyte growth factor + 1% (vol/vol) B27 supplement for 3–6 days. The resultant iTS-P cells were continuously maintained in NSC medium on feeder layers of MMC-treated MEF cells. In the lower panel, photographs show the differentiation process from embryoid bodies into intermediate cells that expressed PDX1 and were capable of producing insulin. We obtained these images from experiments performed exclusively for this review. Scale = 200 μm. ES, embryonic stem; DE, definitive endoderm; GTE, gut tube endoderm; PP, pancreatic progenitors; IPC, insulin-producing cell; PDX1, pancreatic-duodenal homeobox factor-1; EB, embryoid bodies.

**Figure 5 pharmaceutics-13-00780-f005:**
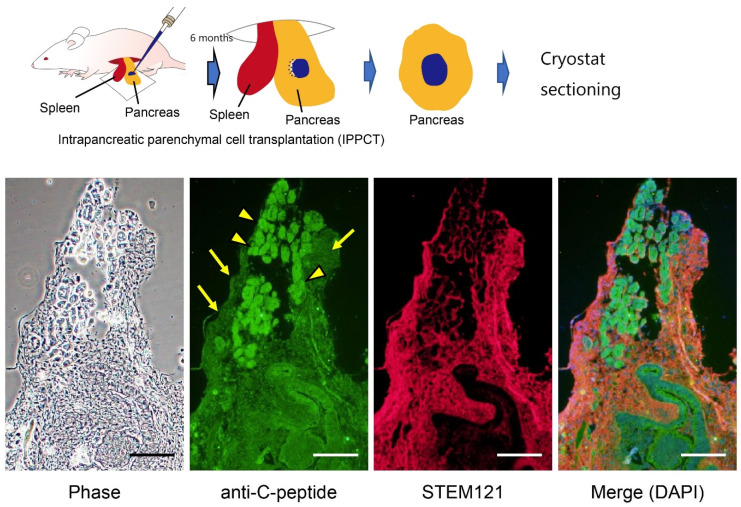
Production of insulin in the grafts after intrapancreatic parenchymal cell transplantation in nude mice. In vitro cultured cell masses (on day 15 after induction; see Figure 4) derived from HDDPC-NSCs were transplanted into the pancreas of nude mice under a dissecting microscope using a glass micropipette (shown above). Six months after grafting, the pancreas was dissected and subjected to cryostat sectioning. Immunostaining using antibodies against insulin (C-peptide) revealed the presence of insulin-positive cells (arrowed) in the graft. Immunostaining using antibodies against STEM121, human cell-specific antibodies, revealed the presence of HDDPC-NSC-derived cells. Arrowheads indicate mouse pancreatic islets. We obtained these images from experiments performed exclusively for this review. Scale = 100 μm.

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
