# Peer review of "Induced Tissue-Specific Stem Cells (iTSCs): Their Generation and Possible Use in Regenerative Medicine"

_pharmaceutics, 2021, doi:10.3390/pharmaceutics13060780_

Round 1

Reviewer 1 Report

In the current review article, Saitoh et al., discussed about the following:

  • The generation of induced pluripotent stem cells (iPSCs)
  • Safety concerns of iPSCs
  • Epigenetic memory associated with iPSCs
  • Induced Tissue-Specific stem cells (iTSCs) as a better alternative.

I find the current review article promising. However, in its current format, the review article fails to demonstrate a significant difference from another published review article (Noguchi et al., 2018). Some of the figures are not clear. Additionally, many paragraphs require rephrasing. I have summarized my concerns below. I highly recommend that the authors address the below mentioned concerns before publishing this review article:

1- The abstract requires rephrasing. In the abstract I find the following opening statement confusing: “ Induced tissue-specific stem cells (iTSCs) are intermediate cells (also progenitors or stem cells) originating during the de-differentiation of differentiated somatic cells into pluripotent stem cells [e.g., induced pluripotent stem cells (iPSCs) and embryonic stem cells (ESCs)]”. It could lead the readers to believe that iTSCs can only be produced as by-products of reprogramming somatic cells to iPSCs. Please rephrase the opening statement to define iTSCs (other than its source) and then mention the various methods to obtain them. After that, discuss briefly about the strategies in the review article.

2-  In the section “Reprogramming of somatic cells into iPSCs” the authors discussed the safety concerns of using iPSCs in therapy due to the usage of viral vectors. I agree with the authors. However, the review in its current format doesn’t discuss other reprogramming strategies (e.g. non-integrating viral vectors, chemical reprogramming… etc). I recommend to briefly discuss the presence of other strategies otherwise the readers might think that retroviral vectors are the only method of reprogramming. Moreover, there are more recent studies regarding the molecular mechanism of reprogramming that the authors can consider citing. Please update this section to make this review article more up-to-date.

3- Figure 1 is very confusing to me. The partial reprogramming as a strategy of obtaining adult stem cells is shown in the figure without being mentioned or introduced in the main text until much later. I recommend separating iTSCs from figure 1 to make it much easier for the readers.

4- In the Epigenetic memory Section, I am against the usage of the term “heat map analysis” (line 121). I recommend removing this statement and just mentioning the findings of the cited manuscript. Heat map analysis oversimplifies the computational strategies used in the cited manuscript.

5- The sentence in lines 166-168 requires rephrasing due to grammatical mistakes.

6- It will be useful if the authors mention whether figure 2 was obtained from experiments performed exclusively for this review article or whether it was obtained with permission from other studies.

7- In line 278, the Hanna et al in-text citation number is wrong.

8- Please rephrase the section “Induction of IPCs from HDDPC-NSCs”. The current format misleads the readers into thinking as if the hypothesis and experiments were done exclusively for this review article. Please make it clearer to the readers that you are explaining a work that is already published (even if it is by the same group that is writing the current review).

9- Figure 3 is not at all clear and requires improvement. The idea that Naive stem cells can better differentiate into iTSCs in comparison with primed stem cells could be explained simply in the text in my opinion. I recommend Figure 3 to become a summary of strategies to obtain iTSCs.

10- Similar to Figure 2, I have the same doubt about Figures 4-5.

11- Rephrasing is required to highlight the strategies that could be used to generate iTSCs (i.e. separate sections for each method). Under each section, the authors can discuss the relevant studies. For example I couldn’t easily find details about the transient OSKM overexpression with additional specific transcription factors. This will help set this current review apart from the similar review article (Noguchi et al., 2018).

Author Response

Reviewer 1

We would like to thank you for your thoughtful consideration of our review, entitled “Induced tissue-specific stem cells (iTSCs): their generation and possible use in regenerative medicine.” We have revised our review according to your comments. All authors have reviewed and approved the changes.

1- The abstract requires rephrasing. In the abstract I find the following opening statement confusing: “ Induced tissue-specific stem cells (iTSCs) are intermediate cells (also progenitors or stem cells) originating during the de-differentiation of differentiated somatic cells into pluripotent stem cells [e.g., induced pluripotent stem cells (iPSCs) and embryonic stem cells (ESCs)]”. It could lead the readers to believe that iTSCs can only be produced as by-products of reprogramming somatic cells to iPSCs. Please rephrase the opening statement to define iTSCs (other than its source) and then mention the various methods to obtain them. After that, discuss briefly about the strategies in the review article.

Response: Thank you for your comment. We rephrased the opening statement about the definition of iTSCs (please see lines 22–25 in the revised text).

2- In the section “Reprogramming of somatic cells into iPSCs” the authors discussed the safety concerns of using iPSCs in therapy due to the usage of viral vectors. I agree with the authors. However, the review in its current format doesn’t discuss other reprogramming strategies (e.g. non-integrating viral vectors, chemical reprogramming… etc). I recommend to briefly discuss the presence of other strategies otherwise the readers might think that retroviral vectors are the only method of reprogramming. Moreover, there are more recent studies regarding the molecular mechanism of reprogramming that the authors can consider citing. Please update this section to make this review article more up-to-date.

Response: Thank you for your comment. We added new information on the reprogramming of somatic cells to iPSCs, as well as recent studies regarding the molecular mechanism of reprogramming (please see lines 115–133 in the revised text).

3- Figure 1 is very confusing to me. The partial reprogramming as a strategy of obtaining adult stem cells is shown in the figure without being mentioned or introduced in the main text until much later. I recommend separating iTSCs from figure 1 to make it much easier for the readers.

Response: Thank you for your comment. We modified Figure 1 in the revised text.

4- In the Epigenetic memory Section, I am against the usage of the term “heat map analysis” (line 121). I recommend removing this statement and just mentioning the findings of the cited manuscript. Heat map analysis oversimplifies the computational strategies used in the cited manuscript.

Response: Thank you for your comment. We modified the indicated portion in the epigenetic memory section of the revised text (please see lines 139–142).

5- The sentence in lines 166-168 requires rephrasing due to grammatical mistakes.

Response: Thank you for your comment. We rephrased this sentence in the revised text (please see lines 188–190).

6- It will be useful if the authors mention whether figure 2 was obtained from experiments performed

Response: Thank you for your comment. We obtained this figure from experiments performed exclusively for this review. We added an explanation to the legend of the new Figure 2.

7- In line 278, the Hanna et al in-text citation number is wrong.

Response: Thank you for your comment. We have corrected this reference.

8- Please rephrase the section “Induction of IPCs from HDDPC-NSCs”. The current format misleads the readers into thinking as if the hypothesis and experiments were done exclusively for this review article. Please make it clearer to the readers that you are explaining a work that is already published (even if it is by the same group that is writing the current review).

Response: Thank you for your comment. We rephrased this section (please see lines 322–325 in the revised text).

9- Figure 3 is not at all clear and requires improvement. The idea that Naive stem cells can better differentiate into iTSCs in comparison with primed stem cells could be explained simply in the text in my opinion. I recommend Figure 3 to become a summary of strategies to obtain iTSCs.

Response: Thank you for your comment. We modified Figure 3 in the revised text.

10- Similar to Figure 2, I have the same doubt about Figures 4-5.

Response: Thank you for your comment. We obtained this figure from experiments performed exclusively for this review. We added an explanation to the legend of the new Figures 4 and 5.

11- Rephrasing is required to highlight the strategies that could be used to generate iTSCs (i.e. separate sections for each method). Under each section, the authors can discuss the relevant studies. For example I couldn’t easily find details about the transient OSKM overexpression with additional specific transcription factors. This will help set this current review apart from the similar review article (Noguchi et al., 2018).

Response: Thank you for your comment. We have added details and rephrased the text. Please see lines 439–450 in particular in the revised text.

Reviewer 2 Report

In this review Saitoh and colleagues critically discuss the pros and cons of iTSCs in comparison to iPSCs and ESCs. The review is well written, I don't have major observations,  however I think that the overall quality of the manuscript will be implemented with few changes. In particular: 

I think the authors should improve the quality of the Figures; in particular I found Figure 1 visually confusing. Thus I suggest them to rethink its organization eventually dividing it in panels. On the other hand, Figure 4 is not well described in the text, since the authors don't discuss about the differentiation protocol. 

Finally I think that the authors should expand the Conclusions describing their viewpoint on the potential future applications of iTSCs. 

Author Response

Reviewer 2

We would like to thank you for your thoughtful consideration of our review entitled “Induced tissue-specific stem cells (iTSCs): their generation and possible use in regenerative medicine.” We have revised our review according to your comments. All authors have reviewed and approved the changes.

In this review Saitoh and colleagues critically discuss the pros and cons of iTSCs in comparison to iPSCs and ESCs. The review is well written, I don't have major observations, however I think that the overall quality of the manuscript will be implemented with few changes. In particular:

I think the authors should improve the quality of the Figures; in particular I found Figure 1 visually confusing. Thus I suggest them to rethink its organization eventually dividing it in panels. On the other hand, Figure 4 is not well described in the text, since the authors don't discuss about the differentiation protocol.

Response: Thank you for your comment. We modified Figure 1 and added more explanation to the legend for Figure 4.

Finally I think that the authors should expand the Conclusions describing their viewpoint on the potential future applications of iTSCs.

Response: Thank you for your comment. We discussed this in the Conclusion section (please see lines 457–460 in the revised text).

Round 2

Reviewer 1 Report

In this round of revision the authors have made significant improvement to the manuscript. The figures are now clearer. The authors have made all the changes which were recommended in a satisfactory manner. The review in its current format is very informative and will be of interest to the readers in my opinion.